# Synthesis of Novel Nano-Sulfonamide Metal-Based Corrosion Inhibitor Surfactants

**DOI:** 10.3390/ma15031146

**Published:** 2022-02-01

**Authors:** Manal M. Khowdiary, Nahla A. Taha, Nashwa M. Saleh, Ahmed A. Elhenawy

**Affiliations:** 1Chemistry Department, Faculty of Applied Science, Umm Al-Qura University Branch El Lieth, Makkah al-Mukarramah 24382, Saudi Arabia; 2Applied Surfactant Laboratory, Egyptian Petroleum Research Institute, Cairo 11727, Egypt; 3Modeling and Simulation Research Department, Advanced Technology and New Materials Research Institute, City of Scientific Research and Technological Applications (SRTA-CITY), New Borg El-Arab City 21934, Egypt; 4Egypt Department of Chemistry, Faculty of Science, Al-Azhar University (Girls Branch), Youssef Abbas Str., Cairo 11651, Egypt; nahlataha.1982@gmail.com; 5Department of Chemistry, Faculty of Science and Arts in Al-Mukhwah, Al-Baha University, Al Bahah 65511, Saudi Arabia; ahmed.elheawy@azhar.edu.eg; 6Chemistry Department, Al-Azhar University, Cairo 11651, Egypt

**Keywords:** critical micelle concentration, cationic surfactant, antitumor activity, corrosion

## Abstract

The synthesis of novel corrosion inhibitors and biocide metal complex nanoparticle surfactants was achieved through the reaction of sulfonamide with selenious acid to produce a quaternary ammonium salt. Platinum and cobalt surfactants were then formed by complexing the first products with platinum (II) or cobalt (II) ions. The surface properties of these surfactants were then investigated, and the free energy of form micelles (ΔG_omic_) and adsorption (ΔG_oads_) was determined. The obtained cationic compounds were evaluated as corrosion inhibitors for carbon steel dissolution in 1N HCl medium. The results of gravimetric and electrochemical measurements showed that the obtained inhibitors were excellent corrosion inhibitors. The anti-sulfate-reducing bacteria activity known to cause corrosion of oil pipes was obtained by the inhibition zone diameter method for the prepared compounds, which were measured against sulfate-reducing bacteria. FTIR spectra, elemental analysis, H1 NMR spectrum, and 13C labeling were performed to ensure the purity of the prepared compounds.

## 1. Introduction

Worldwide, corrosion is one of the greatest problems that educational and industrial divisions must solve because it has a negative effect on the economy of developing and developed countries [1,2]. Corrosion, which results from the presence of a carbon steel in acidic medium, can be reduced by adding corrosion inhibitors in modest amounts [3,4,5,6,7,8,9]. The majority of commercially available acid inhibitors are organic molecules containing heteroatoms, such as sulfur [10], oxygen [11], nitrogen [12,13], and phosphorous. Many heterocyclic compounds containing nitrogen atoms have been found to be excellent inhibitors of steel corrosion in acidic environments [14,15,16,17,18]. Quaternary ammonium salt (QAS) is a famous product utilized in numerous applications of corrosion inhibition of oil pipes. Most of the world’s QAS production involves cationic surfactants, which are widely used on different types of industrial oil protection equipment as well as on closed cooling systems. Synthesis and mass production of these inhibitors have been the primary focus of recent research in science and chemical engineering [19,20,21,22,23]. Biocides are chemical compounds that are capable of preventing or suppressing the growth of bacteria and fungi in different environments. The defeating strategy against lethal microorganisms, especially in the industrial field, involves the use of environmental biocides to protect the environment and living organisms [24,25]. These compounds have other important applications, such as cutting fluids, which are used in various machining operations. Water-miscible cutting fluid has been a recent trend towards water-based fluids. It has the advantage of low toxicity and excellent cooling capacity, and it overcomes problems associated with mineral oil-based products, such as unpleasant odor, oil mist formation, and fumes at high temperatures. However, for water-based cutting fluids to offer a practical alternative, they must possess good rust-inhibiting, anti-wear, and anti-microbial activity [26].

Obviously, in strongly acidic media, most organic inhibitors undergo protonation and exist in their cationic (positive) forms. Conversely, metallic surfaces become negatively charged in acidic solutions due to the adsorption of anionic counter ions of acids (e.g., Cl^−^, SO_4_^2−^, −NO^3−^) [27]. Thus, organic inhibitors initially engage with metallic surfaces through electrostatic attraction forces (i.e., physisorption), and then interact with metallic surfaces utilizing non-binding ions and electrons to form coordination bonds (i.e., chemisorption) [28]. Quaternary ammonium salts have been shown to be exceedingly cost-effective and are frequently employed in the H_2_SO_4_ medium to prevent mild steel corrosion. The efficiency of N-alkyl quaternary ammonium compounds with increasing alkyl chain lengths in reducing acid corrosion of mild steel has been demonstrated [29]. In acid solution, bis quaternary ammonium compounds have also been investigated as corrosion inhibitors [30]. Adsorption is a process in which the inhibitor is absorbed into the body. Adsorption of inhibitors with alkyne moieties causes polymerization of the inhibitors on the metal surface, which results in the creation of a journal pre-proof protective film (coating) and hence, inhibitory effects [31]. Hydrophobic alkyl chains have been shown to boost inhibitory efficiency [32]. Some of these include pyridines, pyrimidines, imidazoles, triazoles, quinolines, naphthyridines, benzothiazoles, benzimidazoles, benzotriazoles, and other similar groups of compounds. The presence of heteroatoms (N, S, O, etc.), bonds, and an aromatic ring structure in these molecules allow them to coordinate with metallic substrates efficiently [33,34]. The purpose of this study was to produce novel compounds that are not hazardous to humans, have a low environmental impact, are biodegradable, and have high inhibition efficiency at low costs. This novel product also may be characterized as a multifunctional material that acts as a biocide, corrosion inhibitor, and emulsifying agent in oil production systems.

## 2. Materials and Methods

All chemicals were produced from Sigma-Aldrich, St. Louis, MO, USA such as selenious acid (H_2_SeO_3_); sulfonamide; ethyl alcohol (C_2_H_5_OH); diethyl ether (C_2_H_5_O); PtCl_2_; CoCl_4_; petroleum ether; cyclodextrin oligosaccharide, carbon steel strips of following composition (by weight, wt.%): C = 0.07, Mn = 0.19, P = 0.02, Si = 0.03, Cr = 0.05, Al = 0.02, Cu = 0.12 and balance Fe.

### 2.1. Preparation of Test Samples

#### 2.1.1. Synthesis of Sulfonamide Hydrogen Selenites IIa

To carry out the synthesis process, stoichiometric amounts of selenious acid were mixed with sulfonamide in diethyl ether at 25 °C with vigorous shaking until precipitation was complete. Then, products were filtered and washed with acetone and were recrystallized by petroleum ether [35]. The products were designated as IIa and have the general formula:RNH_3_ HSeO_3_
where R = sulfonamide

#### 2.1.2. Synthesis of Metal Complexes


(a)Synthesis of cobalt and platinum hydrogen selenite dehydrate [35].
PtCl_2_·2H_2_O + Na_2_CO_3_ → PtCO_3_·2H_2_O + 2NaCl
CoCl_2_·2H_2_O + Na_2_CO_3_ → CoCO_3_·2H_2_O + 2NaCl
PtCO_3_·2H_2_O + 2H_2_SeO_3_ → Pt(HSeO_3_)_2_ +2H_2_O
CoCO_3_·2H_2_O + 2H_2_SeO_3_ → Co(HSeO_3_)_2_ + 2H_2_O(b)Synthesis of platinum and cobalt ammonium hydrogen selenite complexes IIb,c.


Platinum or cobaltsulfonammonium hydrogen selenite complexes were prepared by refluxing two moles of sulfonammonium hydrogen selenites (IIa) with one mole of platinum or cobalt hydrogen selenite in ethyl alcohol for two hours. The products were designated as (IIb,c), as shown in Figure 1.
2RNH_3_ (HSeO_3_)_2_ + M (HSeO_3_)_4_ → (RNH_3_)_2_M(HSeO_3_)_4_^−^
where
M = Pt or Co  R = sulfon

Compound IIb, Yield, 72%; m.p. 172–174 _C; IR, (KBr, cm^−1^): disappear of NH_2_ according to form quaternary salt, 3053 (CH arom.), 1370, 1155 (SO_2_). 1H-NMR (DMSO-d6) δ: 4.6 (s, 1H, NH, exchangeable with D_2_O), 7.3–8.0 (m, 10H, Ar-H + SO_2_NH_2_, exchangeable with D_2_O). MS m/z (%): 1056 (57.77), 172 (100). Anal. Calcd. For C_12_H_22_N_4_O_16_S_2_PtSe_4_ (1056.68): C, 13.68; H, 2.11; N, 5.32; O, 24.30; Pt, 18.52; S, 6.09; Se, 29.98. Found: C, 13.56; H, 2.08; N, 5.29; O, 24.24; Pt, 18.42; S, 5.99; Se, 29.78.

Compound IIc, Yield, 82%; m.p. 210-212 _C; IR, (KBr, cm^−1^): disappear of NH_2_ according to form quaternary salt, 3052 (CH arom.), 1373, 1156 (SO_2_). 1H-NMR (DMSO-d6) δ: 4.5 (s, 1H, NH, exchangeable with D_2_O), 7.6–8.3 (m, 10H, Ar-H + SO_2_NH_2_, exchangeable with D_2_O). MS m/z (%): 920 (37.16), 172 (100). Anal. Calcd. For C_12_H_22_N_4_O_16_S_2_PtSe4 (920.65): C, 15.71; H, 2.42; Co, 6.43; N, 6.11; O, 27.91; S, 6.99; Se, 34.43. Found: C, 15.69; H, 2.32; Co, 6.39; N, 6.09; O, 27.83; S, 6.86; Se, 34.37.

#### 2.1.3. General Formula for the Metal Complexes

Metal complex prepared using either platinum or cobalt ([RNH_3_] + 2Pt[(HSeO_3_)^−4^], [RNH_3_] + 2Co[(HSeO_3_)^−^]_4_ ) may be formulated as expected with the chemical structure shown in (Figure 2)

#### 2.1.4. Green Synthesis with Solid State Reaction in Ball Mill for Complex Nanoparticles

The method of synthesizing organic compounds using a ball mill as a reactor is an environmentally safe method for using green chemistry. By comparing this technique with normal solution methods, it presents many advantages, such as reducing the number of solvents used, thereby reducing the number of volatile compounds. Heavy mixing in the solid state helps disintegration and mobility for rapid friction among molecules in a short amount of time. For non-polar and polar solvents, mixing problems are reduced. The energy used to complete the reaction is reduced due to the optimum use of energy in the reaction mixture. The nano-sized particles of Co and Pt complexes were achieved by mixing them very well with cyclodextrin oligosaccharides using ceramic mortar. Finally, both were ground using a ball mill model PM 200 (TMAXC-Fujian, China) at 300 rpm for 3 h.

#### 2.1.5. Evaluation of Anti-Sulfate-Reducing Bacteria Activity

The inhibition zone diameter (Iz D) Method: a filter paper sterilized disc saturated with a measured quantity of the sample was placed on a plate containing a solid bacterial medium (nutrient agar broth) that had been heavily seeded with the spore suspension of the tested bacteria. After incubation, the diameter of the clear zone of inhibition surrounding the sample was taken as a measure of the inhibitory power of the sample against the particular test organism. All sulfate-reducing bacteria were provided by a culture collection of the Regional Center for Mycology and Biotechnology (RCMB), Al-Azhar University, Cairo, Egypt.

### 2.2. Corrosion Inhibition Measurements

#### 2.2.1. Weight Loss Measurements

The carbon steel specimens have a composition of (wt %): 0.21 C, 0.035 Si, 0.25 Mn, 0.082 P, and the remainder is Fe. The carbon steel sheets of 2.5 cm × 2.0 cm × 0.6 cm were abraded with emery papers (grades 320, 500, 800 and 1200) and then washed with distilled water and acetone. After weighing accurately, the specimens were immersed in a 250 mL beaker containing 200 mL of 0.5 M hydrochloric acid alone or with 50, 100, 200, or 400 ppm by weight of the inhibitors used at 25 °C. After different immersion time intervals of 1, 3, 6, and 24 h, the specimens were taken out, washed, dried, and weighed accurately. The corrosion rate (K) and the inhibition efficiency (*η* %) were calculated using the following equations [36].
K = (W/St)
η_w_ % = [{W_corr_ − W_corr(inh)_}/W_corr_] × 100
where W is the average weight loss of three parallel carbon steel sheets (one specimen in each beaker), S is the total area of the steel specimen, and t is immersion time. W_corr_ and W_corr(inh)_ are the corrosion rates obtained in the absence and the presence of inhibitors, respectively.

#### 2.2.2. Polarization Measurements

Electrochemical measurements are made in the traditional cylindrical cells made of Pyrex glass with a triple electrode, the working electrode (WE) in the form of steel rods embedded with polytetrafluoroethylene (PTFE; the Free Zone of the electrode that was exposed to the analysis of the electrical 0.7 cm^2^) and saturated calomel electrode (SCE), a platinum electrode used as auxiliary, and a standard electrode, respectively. Polarization measurements were carried out using a potentiostat (Wenking, Wuhan, China). The working electrode was immersed in the test solution for 45 min to reach the open circuit potential. (The carbon steel used in the polarization measurements was identical to that used in the weight loss measurements.) Next, the working electrode was polarized in both cathodic and anodic directions. In terms of accuracy of the calculated slopes, the values were compared with the obtained data from the software calculations accompanied with a potentiostat (Wenking, Wuhan, China), and the accuracy was 4%. The values of corrosion current density (I_Corr_) were calculated via extrapolation of the straight part of the Tafel lines. A standard ASTM glass electrochemical cell was used, and the platinum electrode was used as an auxiliary electrode. The potential increased with a speed of 2 mV min^−1^, starting from −200 mV to +200 mV with respect to the open circuit potential (OCP) versus corrosion potential (E_Corr_) [36].

#### 2.2.3. Scanning Electron Microscopy (SEM)

After dipping carbon steel samples in acidic solution, corrosion of the samples was tested with different concentrations of the corrosion inhibitor products to investigate the optimal concentration of each inhibitor at which the corrosion decreased. Exposing the samples for SEM also directly aided in studying the surface’s morphology.

## 3. Results and Discussion

### 3.1. Surface Properties of the Prepared Cationic Surfactants

As shown in Table 1 and Figure 3, after complexing ammonium hydrogen selenite with Co or Pt ions, a high depression in CMC, Y_cmc_, A_min_ value was observed for compounds designated as IIa, IIb and IIc compared to those of the complex designated as IIa. This may be because metal complexes have specific properties in water. The complexes in the solutions aggregate as units, which can increase the volume due to repulsion between the hydrophobic chain of the complexes and water molecules that can occur.

An increase in Pc20, Π_CMC_, and Γ_max_ value was observed for compounds designated as IIb and IIc in comparison to the complex designated as IIa. This may be attributed to the double chain of hydrocarbon groups in metal complexes, which by its role, increases the hydrophobicity, thereby causing an increase in the concentration at the interface (maximum surface excess Γ_max_). Froing of the great number of molecules at the interface leads to an increase in the effectiveness of Π_CMC_ and a greater reduction in surface tension at critical micelle concentration [37]. All the result values of ΔG_ads_ and ΔG_mic_ were negative, which is considered evidence for carrying them out directly without increasing the temperature of the reaction system.

### 3.2. Antibacterial Activity of the Prepared Surfactants against Sulfate-Reducing Bacteria

#### Corrosion Inhibitor Activity for Oil Pipes

Sulfate-reducing bacteria (SRB) are considered reducers of sulfate, and their growth and reproduction cause oil pipe and equipment corrosion. This can lead to the loss of economy, and cause environmental, health, and safety hazards. Stabilized mixed culture accumulates with sulfate reducing bacteria. In many industrial sectors such as the oil and gas industry, it is important to minimize the evolution of hydrogen sulfide gas which can result from SRB activity. The results of the synthesized cationic surfactants against sulfur-reducing bacteria are recorded in Table 2.

As shown from Table 2, the novel products have high antibiological activity against SRB, while this activity also increased as we notice from the table. When products form complexes that have two hydrocarbon groups, hydrophobicity adsorption at the bacterial membrane increases. The protein and lipid molecules of the complex mobilize with the hydrocarbons of the inhibitor and the membrane breaks down, inhibiting the growth of DNA and thus no reproduction. Platinum complexes produced the best results, possibly because platinum is an oxidizing chemical agent that acts as a reduction inhibitor, reducing sulfide generation and slowing anaerobic growth (SRB). The specific activity of the tested bacteria was closely related to the rate of H_2_S generation and the generation time was inversely proportional to these activities. The rate of bacterial resistance to metal ions was directly proportional to the rate of microbiologically induced corrosion (MIC) of carbon steel.

### 3.3. Results of Weight Loss (Gravimeteric) Method and Effect of Inhibitor Concentration

Table 3 and Table 4 describe the variations in corrosion rate (K values) of the dissolution reaction of carbon steel in 1 M HCl solution in the presence of IIb and IIc inhibitors at different concentrations, respectively. It is obvious that a gradual increase in inhibitor concentrations from 1 × 10^−4^ to 1 × 10^−2^ M by weight reduces the K values considerably. At 1 × 10^−4^ M by weight of the II_b_ and IIc inhibitors, the K values of carbon steel were high. In contrast, increasing the inhibitor concentration to 1 × 10^−2^ M by weight decreases K to the lowest value. This is due to an increase in the number of adsorbed inhibitor molecules on the metal surface caused by an increase in the inhibitor concentration in the medium. In turn, this improves surface protection and slows the metal dissolution reaction. The number of inhibitor molecules adsorbed on the metal surface increases as the inhibitor concentration increases. As a result, this increases the surface area covered by inhibitor molecules and consequently decreases the dissolution of metal due to the reaction of metal by corrosive ions in the medium [37].

### 3.4. Electrochemical Evaluation

#### Potentiodynamic Polarization Spectroscopy

Cathodic and anodic current for carbon steel corrosion in 1 M HCl in the presence of 1 × 10^−2^ to 1 × 10^−4^ M by weight of the synthesized inhibitors II_b_, IIc at 25 °C. Figure 4 represents the relation for (IIb) where (βc, βa): the electrochemical polarization parameter of cathodic and anodic Tafel slopes; (I_corr_): corrosion current density; (E_corr_): potential of corrosion; (η): inhibition efficiency obtained from this figure were listed in Table 5. The corrosion currents decreased in the presence of the inhibitors as indicated by low values, such as 0.06, 0.04, 0.02, and 0.24 mA cm^−2^ for 1 × 10−4 to 1 × 10^−2^ M by weight of II b respectively. Surface coverage (θ) of the carbon steel by the inhibitor molecules was determined by using the values of the corrosion currents in the following equation [38]:θ = (i_corr(uninh)_ − I_corr(inh)_)/(I_corr(uninh)_)
where I_corr (uninh)_ is the corrosion current density with the presence of different inhibitors and I_corr(inh)_ is the corrosion current density in the absence of the different inhibitors.

Corrosion inhibition efficiencies can be calculated from the previous values with different concentrations of the inhibitors using the following equation [39]:η% =(I_corr(uninh)_ − I_corr(inh)_)/(I_corr(uninh)_) × 100

As shown in Table 5, at 1 × 10^−2^ M by weight, the value of i_corr_ in the case of IIc (0.020 mA cm^−2^) is smaller than that of II_b_, while accordingly, the values of inhibition efficiencies in the presence of IIc (94.68) is greater than that of IIb (83.01%). From the polarization curves displayed in Figure 4, the decrease in the two reactions at the cathode and anode occurs in the absence of the synthesized corrosion inhibitors but not in the presence of them. Additionally, increasing the concentration from 1 × 10^−4^ M to 1 × 10^−2^ M by weight increases inhibition. It may be that the presence of inhibitors in higher concentrations reduced the anodic dissolution and suppressed the reduction of hydrogen ions [40,41]. As shown in Figure 4, by increasing the concentration of the synthesized product, the cathodic current densities sharply decreased, which may be due to the formation of a covered separation film which developed from a single layer of the synthesized product onto the cathodic sites of the carbon steel [42]. The corrosion current densities increased slowly at the start of anodic polarization, and the anodic polarization was mostly facilitated by the positive shift in polarization potential. This indicates that the adsorption rate of the inhibitor molecules on the carbon steel surface is greater than their desorption rate and that the adsorption process controls the anodic reaction (as in Figure 5). The polarization behavior was comparable, implying that the inhibitor molecules are fully adsorbed from the solution to the carbon steel surface and that increasing the concentration increases their inhibition action [43]. When the inhibitor concentration increases, corrosion decreases. With increasing inhibitor concentrations, the protective coating adsorbed onto the metal surface tends to be more complete. This is supported by the increase in corrosion current densities and the increase in η%. When the corrosion potential shifts to values greater than 85 mV (compared to the corrosion potential without inhibitors), the inhibitors are categorized as cathodic or anodic [44]. However, for all the inhibitors tested, the E_corr_ displacement is less than 85 mV. As a result, these inhibitors are known as mixed-type inhibitors [45,46].

### 3.5. Docking Study

Docking investigations based on potent–protein interactions are used to explain the biological results. Figure 6 shows all of the docking computations. Protein data library was used to obtain the X-ray crystal structures of the following bacterial strains: A. fumigatus (ID [47]; 5HWC), G. Candidum (ID [48]; 4ZZt), S. Pneumoniae (ID [49]; 5LJI), and E. Coli (ID [50]; 3t88), respectively. The following are the essential backbone amino acid residues retrieved via X-rays analysis: Tyr27, Asp38, Arg55, Ile57, and Tyr83 for 5HWC; Arg339, Arg251, Glu212, and Asp259 for 4ZZW; Gly 60, Glu97, Cys99, Asp92, Thr15, Asn14, Ser90, and Thr58 for 5LJI; Tyr57, Asn42, Thr203, Arg188, and Glu213 for 3t88. These amino acid residues hindered microbial action through chelating with reference inhibitors. Ag/TNTs-P400 was the most effective at docking into kinase receptors. MOE-scoring for the most stable docking model was used to evaluate the binding affinity of complexes (inhibitor-kinase) (Figure 6). With an MMFF94 force field, the complexes were energy-minimized until the gradient for the restricted convex was minimized to 0.05 kcal/mol. IIb was the most effective at docking onto kinase receptors. MOE-scoring was used to identify the most stable docking model used to evaluate the binding affinity of complexes (inhibitor-kinase) (Figure 6). For all kinases, the studied sample was successfully docked onto active sites. In Figure 6, the hydrogen bond interactions between the ligand and the receptor were discovered in the construction of active nanocomposites. The hydrophobic cation-π interactions at a distance >6 Å are not shown in Figure 7. In the absence of H-bond interactions, IIa docked onto the active sites 5HWC, 4ZZt, 5LJI, and 3t88, which gave insignificant binding energies (Figure 7). The sample has the greatest binding score (G =~ −4 Kcal/mol) against the 4ZZt active site.

The kinase reset inhibition potency assumed the following pattern: 5LJI>3t88 >5HWC, with promising binding score values (Figure 7). Amino acid residues interact with samples by forming significant E.H.B of H-bonds with energies ranging from ~ −8 to −5 kcal/mol. Against the 4ZZt binding pocket, the studied materials had the highest stabilization energy (E_int_ = −8.267 kcal/mol). On the other hand, those found at the 3t88 active site had the lowest stabilization energy (E_int_ = −2.140 kcal/mol). In comparison to 5LJI binding pockets, 5HWC binding pockets yielded less stabilization energy (Figure 7).

## 4. Conclusions

This work aimed to synthesize novel antibiotic quaternary ammonium salt nano-surfactants based on sulfonamide with selenious acid and complexed with platinum and cobalt surfactant. We used FTIR spectra, elemental analysis, H1 NMR spectrum, and 13C labeling to investigate the surface behaviors of these surfactants as free energy to form micelles (ΔG_omic_) and adsorption (ΔG_oads_). In 1N HCl media, we assessed the generated cationic compounds as corrosion inhibitors for carbon steel. Gravimetric and electrochemical studies revealed that the studied inhibitors are excellent corrosion inhibitors. The anti-sulfate-reducing bacteria activity, which causes the corrosion of oil pipes, was obtained by the inhibition zone diameter method for the prepared compounds, which measured against sulfate-reducing bacteria. Particle size scaled into the nano-range, which required investigation via transmitted electron microscope (TEM). Finally, the DFT using B3LYP and the 6-311G* correlation function was applied for the investigated probes. DFT and MEPs were also used to identify the interaction compound’s behavior over heavy metal for water formation.

## Figures and Tables

**Figure 1 materials-15-01146-f001:**
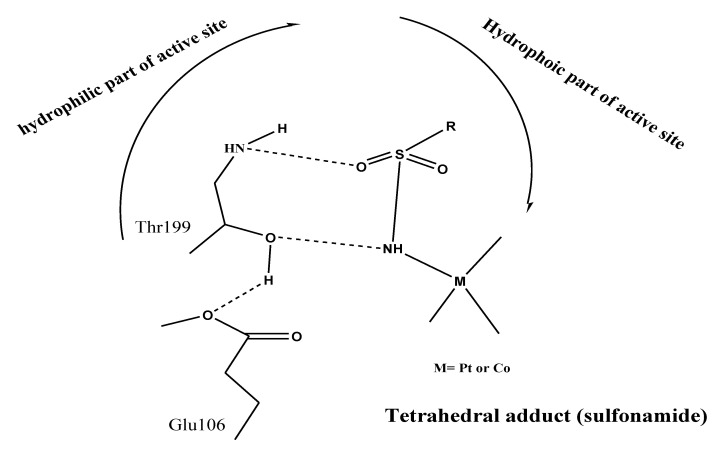
CA inhibition mechanism by sulfonamides.

**Figure 2 materials-15-01146-f002:**
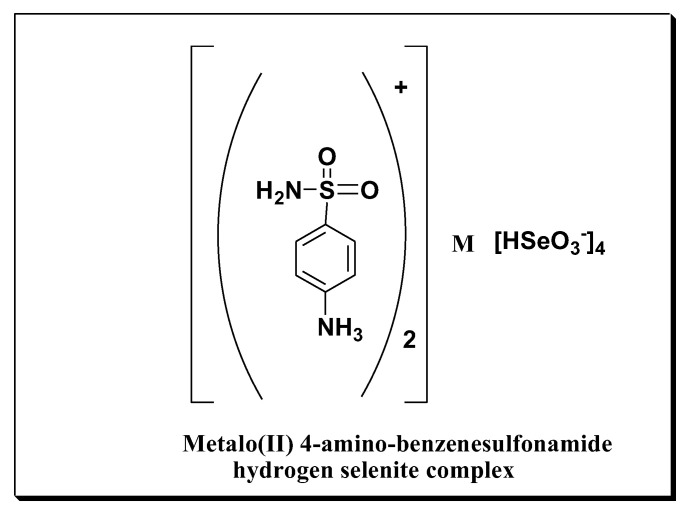
Metalo (II) 4-amino-benzenesulfonamide hydrogen complex.

**Figure 3 materials-15-01146-f003:**
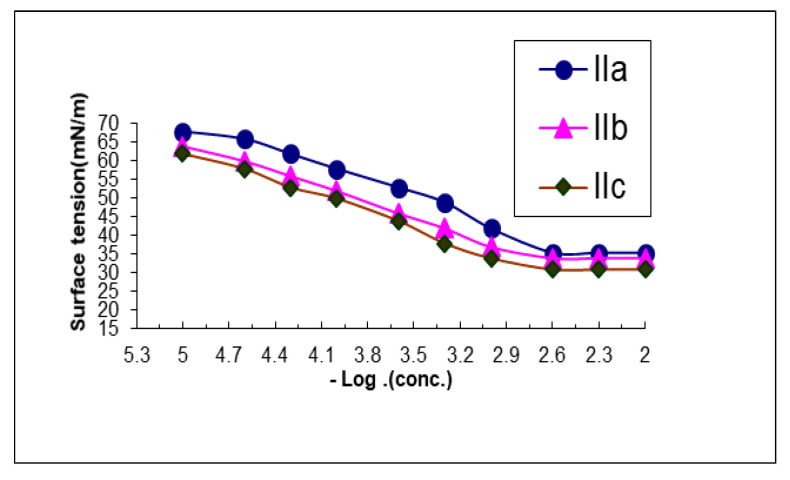
Variation in surface tension of surfactants IIa-c vs. concentration at 25 °C concentration expressed as mol/L.

**Figure 4 materials-15-01146-f004:**
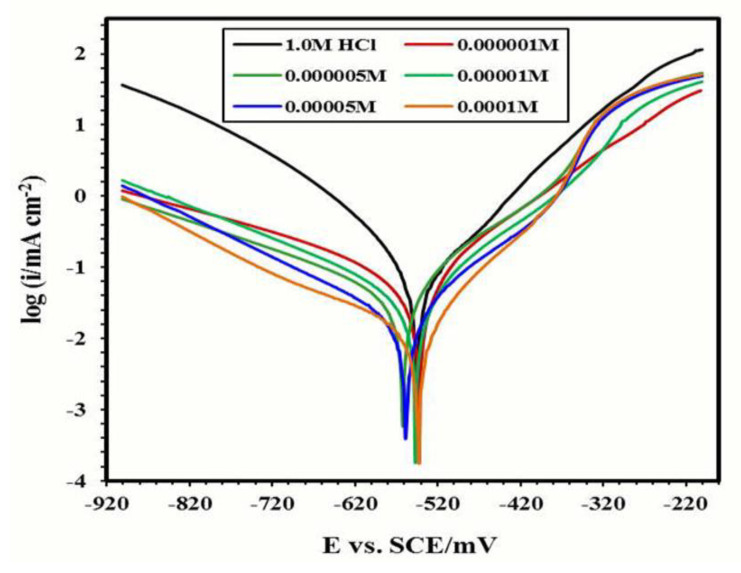
Polarization curves of carbon steel corrosion in 1 N HCl solution containing different concentrations of II_b_ inhibitor at 25 °C.

**Figure 5 materials-15-01146-f005:**
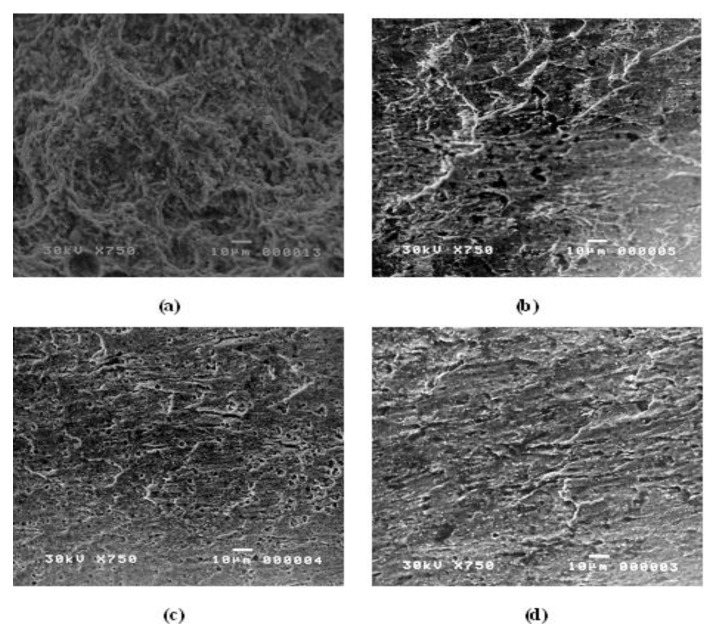
Image of the surface appearance of immersed samples in 1 M HCl by Scanning Electron Microscope (**a**) without inhibitor; and (**b**–**d**) in the presence of 1 × 10^−1^ 10, 1 × 10^−2^, 1 × 10^−3^ M IIb inhibitor, respectively.

**Figure 6 materials-15-01146-f006:**
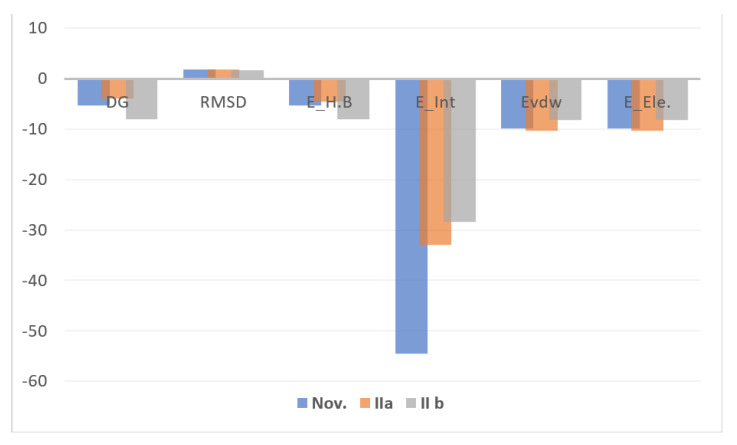
Docking energy scores (kcal/mol) for investigated compounds. ΔG: Free binding energy of the ligand from a given conformer; Int.: Affinity binding energy of hydrogen bond interaction with receptor; H.B.: Hydrogen bonding energy between protein and ligand; Eele: Electrostatic interaction with the receptor; Evdw: Van der Waals energies.

**Figure 7 materials-15-01146-f007:**
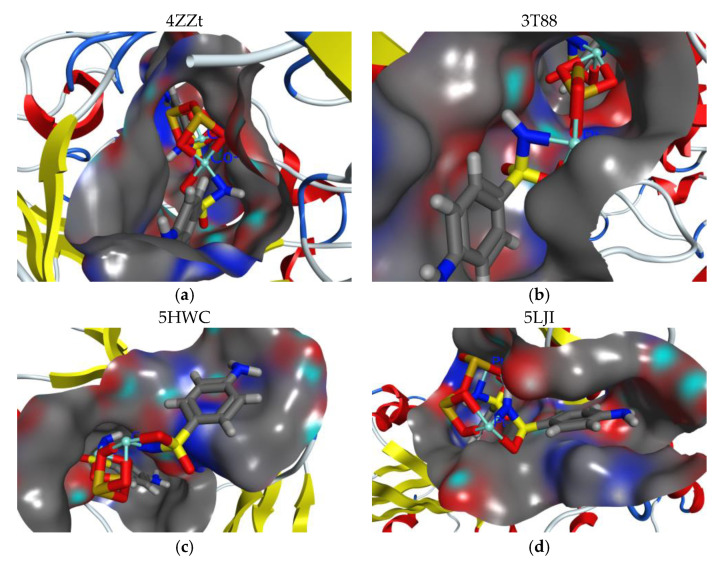
The Docked most active complexes into the electrostatic surface active sites of IDs: 4ZZt, 3T88, 5HWC and 5LJI using MOE tool [32].

**Table 1 materials-15-01146-t001:** The critical micelle concentration (CMC) and surface parameters of synthesized surfactants.

Comp. No.	CMC X 10^−3^	Γ_cmc_ (mN/m)	Π_cmc_ (mN/m)	PC20 (Mole/L)	Γ_max_ X 10^−11^ (Mole/cm^2^)	A_min_ (nm^2^)	Δ G_ads_	Δ G_mic_	ΔG_ads_/A_min_
II_a_	1.2	32	40	3.9	10.4	1.5	−67.7	−34.1	−46.8
II_b_	1.1	30	42	4.1	10.2	1.5	−69.9	−34.8	−49.1
II_c_	0.80	29	43	4.3	11.1	1.45	−71.1	−35.3	−50.2

**Table 2 materials-15-01146-t002:** Inhibition zone diameter (mm/mg sample) for the synthesized cationic surfactants against sulfate-reducing bacteria.

Sample	Inhibition Zone Diameter (Iz D) (mm/mg Sample) Sulfate-Reducing Bacteria
IIa	22
IIb	20
IIc	18

**Table 3 materials-15-01146-t003:** Effect of cationic surfactant inhibitor concentration (IIb) at 30 °C temperature degrees.

Temperature °C	Conc. of Inhibitor M	*K* mg cm^−2^ h^−2^	*η*w %
30	0.00	1.307	-
	1 × 10^−4^	0.838	35.86
	5 × 10^−4^	0.643	50.80
	1 × 10^−3^	0.402	69.26
	5 × 10^−3^	0.298	77.16
	1 × 10^−2^	0.241	81.54

**Table 4 materials-15-01146-t004:** Effect of cationic surfactant inhibitor concentration (IIc) at 30 °C temperature degrees.

Temperature °C	Conc. of Inhibitor M	*K* mg cm^−2^ h^−2^	*η*w %
30	0.00	1.4	-
	1 × 10^−4^	0.5	61.82
	5 × 10^−4^	0.261	80.01
	1 × 10^−3^	0.222	83.01
	5 × 10^−3^	0.103	92.11
	1 × 10^−2^	0.070	94.68

**Table 5 materials-15-01146-t005:** Electrochemical polarization parameters of carbon steel corrosion in the presence different concentrations of II_b_ and IIc inhibitors.

Inhibitor Name	Conc. of Inhibitor (M)	E_corr_ (mV)	I_corr_ (mAcm^−2^)	β^a^ (mV/Decade)	β^c^ (mV/Decade)	θ	η_P_%
Without inhibitor	0.00	−487.3	2.02	208.1	202.4	-	-
II_b_	1 × 10^−4^	−499.0	0.0646	118.0	−137.4	0.72	71.55
	5 × 10^−4^	−485.5	0.0345	114.8	−161.6	0.85	84.81
	1 × 10^−3^	−485.4	0.0299	137.2	−194.2	0.87	86.83
	5 × 10^−3^	−477.2	0.0280	117.6	−104.2	0.88	87.67
	1 × 10^−2^	−504.7	0.0243	128.1	−140.3	0.89	89.30
II_c_	1 × 10^−4^	−563.6	0.0530	116.5	−176.3	0.77	76.66
	5 × 10^−4^	−475.7	0.0315	128.2	−112.7	0.86	86.13
	1 × 10^−3^	−478.8	0.0277	127.6	−106.3	0.88	87.80
	5 × 10^−3^	−509.6	0.0244	108.8	−170.1	0.89	89.26
	1 × 10^−2^	−496.9	0.0233	125.8	−196.5	0.90	89.74

## Data Availability

The data presented in this study are contained within the article.

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
