# Peer review of "Synthesis of Novel Nano-Sulfonamide Metal-Based Corrosion Inhibitor Surfactants"

_materials, 2022, doi:10.3390/ma15031146_

Round 1

Reviewer 1 Report

The work is devoted to the important issue of finding ways to slow down the corrosion of steel.

First of all, I recommend to authors to review the work and make changes based on the guidelines written below, it must be rejected.

  1. It is necessary to clearly distinguish between introduction, materials and methods, preparation of test samples, results and discussion. Please read your article again and check for commas, spaces, periods, etc. There is a huge number of punctuation errors.
  2. Line 24, 316, and Line 255: It is not clear in which corrosive electrolyte the investigations were carried out.
  3. The introduction does not indicate the purpose of the work. It is not clear why everything was done.
  4. Line 74: The authors must indicate all chemical compounds used in the article with chemical formulas in brackets.
  5. Line 117: The authors must indicate the type of jar used, the material and mass of the prayer bodies.
  6. Line 123: The application of the laser particle size analyzer (FRITSCH) model Analystte 22 is described. I do not see graphs and tables of the results obtained. Where the TEM picture came from, the materials and methods do not indicate TEM use as a particle size analysis. It is necessary to add a description of the microscope or remove picture 3. It is necessary to increase the scale of the data labels and the ruler in Figure 3.
  7. Points 2.1.4 and 2.1.5 must be combined because they have one topic for describing ball milling.
  8. Point 2.2.1 - 2.2.2: It is not permissible to refer to the source of literature without explaining the data obtained.
  9. Point 3.1: The theories put forward must be substantiated in more detail and refer to similar works.
  10. In the materials and methods, there aren’t the parameters of corrosion studies, the used electrochemical cell, the type of electrodes, scanning parameters, the composition of the electrolyte, etc.
  11. In materials and methods, there isn’t the chemical composition of the steel to which the inhibitor was applied.
  12. Line 229: There aren’t any details of the corrosion processes and reactions occurring in a corrosive environment. The potentials of these reactions must be indicated.
  13. Line 255: To compare corrosive behavior, a graph of carbon steel without inhibitors should be presented. Table 5 shows 11 results of the study and in Figure 5 there are only 6 results. The required curves must be added. Here is needed to expand the discussion, to compare the results obtained with other authors.
  14. Figure 6 The morphology of the coating is not clear !!! There isn’t any description of the resulting morphology, which microscope was used ??? Where is the chemical analysis of the surface? How can I tell if an inhibitor was actually present on the surface of the samples?
  15. Potentiodynamic curves are not enough to characterize corrosion processes and substantiate a good inhibiting ability; impedance spectroscopy must be done.
  16. Line 313: In this sentence, the methods of analysis used in this article are listed, I did not see the results and graphs.

Author Response

added in the attached file 

Reviewer 2 Report

The manuscript can be accepted after major revision: 1. The introduction should be revised carefully to show the novelty of this work. Moreover, the similarity index is 28% which needs to be reduced. I discovered that there are sections that seem to be unoriginal, having appeared in previously published work(s). In this case, the overlap goes beyond the normal occurrence of standard phrases in this field. 2. All figures have poor qualities. For example, Fig.6 3. The discussion of polarization curves analysis was not sufficiently discussed. Moreover, the unit of Tafel slopes in Table 5 should be mV/decade. Please revise it. 4. Please check out all units, subscripts, and superscripts in the manuscript. 3. The authors ought to revise the manuscript again to eliminate grammatical and spelling errors. In addition, the Results and discussion part should need to be rewritten to avoid confusion and ambiguity

Author Response

added in the attached file 

Round 2

Reviewer 1 Report

I propose to accept the manuscript

Reviewer 2 Report

Accept in present form